# Mechanical and Microstructural Characterization of Arc-Welded Inconel 625 Alloy

**DOI:** 10.3390/ma12223690

**Published:** 2019-11-08

**Authors:** Daniel Kotzem, Lucas Beermann, Mustafa Awd, Frank Walther

**Affiliations:** Department of Materials Test Engineering (WPT), TU Dortmund University, Baroper Str. 303, D-44227 Dortmund, Germany; lucas.beermann@tu-dortmund.de (L.B.); mustafa.awd@tu-dortmund.de (M.A.); frank.walther@tu-dortmund.de (F.W.)

**Keywords:** additive manufacturing, arc welding, mechanical behavior, high-temperature alloy

## Abstract

The objective of this work was to verify a relatively new fusion-based additive manufacturing (AM) process to produce a high-temperature aerospace material. The nickel-based superalloy Inconel 625 (IN625) was manufactured by an arc-based AM technique. Regarding microstructure, typical columnar-oriented dendritic structure along the building direction was present, and epitaxial growth was visible. The mechanical behavior was characterized by a combination of quasi-static tensile and compression tests, whereas IN625 showed high yield and ultimate tensile strength with a maximum fracture strain of almost 68%. Even quasi-static compression tests at room and elevated temperatures (650 °C) showed that compression strength only slightly decreased with increasing temperature, demonstrating the good high-temperature properties of IN625 and opening new possibilities for the implementation of arc-based IN625 in future industrial applications.

## 1. Introduction

The Ni-based superalloy Inconel 625 (IN625) is frequently used, especially in the aerospace and automotive sector, due to its unique mechanical properties even under high temperatures and in corrosive environments [1,2,3]. Although huge efforts have been made to use traditional manufacturing methods in order to produce components with acceptable mechanical properties, high production costs and complex geometries are still restrictive factors to reaching a wide scope of applications [4]. In the case of IN625, additive manufacturing (AM) techniques open a new research area in comparison to traditional manufacturing methods. AM processes include selective laser melting (SLM) [5], electron beam melting (EBM) [6,7], and laser metal deposition (LMD) [8].

In comparison to the aforementioned manufacturing techniques, arc-based AM techniques have received increasing attention due to significant cost savings, high deposition rates, and simple handling [9,10]. So far, arc-welding processes are more commonly used for joining different components, especially tailored blanks for lightweight automobile bodies [11], or applying coatings [12]. The processability of building 3D parts by arc-based welding has already been developed by various researchers in the 1990s [13,14,15]. In general, the accuracy and surface roughness of arc-based AM processes cannot reach the level of laser and electron beam processes, but nevertheless, arc-based AM processes are promising new technologies for the production of future large-scale IN625 components.

Baufeld [10] investigated Inconel 718 (IN718) parts manufactured by gas tungsten arc welding. There, especially microstructural and mechanical characterization were carried out, and related results showed that arc-based AM processes can be used for the manufacturing of near-net-shaped dense IN718 components. In comparison to as-cast material, slightly higher tensile properties were reached due to lower cooling rates in comparison to laser or electron beam techniques. Further on, Baufeld presumed that additional heat treatments could further increase the tensile strength. Wang et al. [4] investigated the effect of location on the microstructure and mechanical properties of IN625 components also manufactured by gas tungsten arc welding. Even they successfully achieved the fabrication of fully dense components with no relevant defects, like cracks or pores. Wang et al. further determined mechanical heterogeneity along the vertical building direction based on a varying temperature gradient. Related results showed higher tensile strength and an increased micro-hardness in the top region due to the heterogeneous microstructure, large dendrite arm spacing, and an increased number of Laves phases.

To the best of the authors’ knowledge, only a few papers can presently be found in the literature about the microstructural and mechanical characterization of IN625, especially under operating temperatures (T > 0.4 T_S_), produced by arc-based AM processes. Therefore, the present work focusses on arc-welded IN625, which is characterized by optical metallography, scanning electron microscopy (SEM), defect distribution, and hardness measurements. Further on, a combination of quasi-static tensile and compressive tests at room and high temperatures were carried out in order to investigate the effect of loading orientation and temperature on the local mechanical properties of the fabricated material.

## 2. Materials and Methods 

The investigated IN625 material was manufactured by arc welding and was supplied by Technical University Ilmenau (Ilmenau, Germany). Commercially available IN625 welding wire was deposited in the building direction (Z) onto the substrate layer by layer. Over 50 deposited layers were manufactured, resulting in an upright standing plate with final dimensions of approximately 200 × 120 × 11 mm³, as depicted in Figure 1. The deposition direction was along the X-axis and identical to that of the previously deposited layer. Further details regarding process parameters are not available.

For metallographic and mechanical characterization, the used specimens were extracted from the aforementioned component (Figure 1). For light microscopy, the specimens were cold-embedded in epoxy resin, ground with abrasive paper, and finally polished with a diamond suspension (3 µm and 1 µm). Afterward, the samples were etched with the Beraha II etching method for 360 s. Light microscope images were taken in the horizontal (X–Y plane), longitudinal (X–Z plane), and transverse (Y–Z plane) directions with an Axio Imager light microscope (Carl Zeiss, Oberkochen, Germany). For microstructural investigations at higher magnifications up to 3000× and for fractographic analysis of the tensile specimens the SEM MIRA3 XMU (Tescan, Brno, Czech Republic) was used.

Due to the high density of IN625, penetrating the complete specimen with µ-CT was difficult. Therefore, the relative density was examined by acquiring light microscope images and performing an image analysis using the dhs-Bilddatenbank^®^ software (dhs Dietermann & Heuser Solution, Greifenstein-Beilstein, Germany). Five images were taken for each sample. In total, two samples were investigated, and an average relative density was determined. From that, a total surface area of 120.9 mm^2^ was analyzed.

Macro-hardness was measured on a Wolpert Dia-Testor 2Rc Vickers hardness-testing machine (Instron^®^, Norwood, MA, USA) with a static load of 98.06 N (HV10) and a dwell time of 12 s. The indentations were made along the building direction (Z) in the Y–Z plane with intervals of 2 mm. For every build-height, at least five hardness measurements were carried out, and values of average hardness were determined according to ISO 6507 [16].

The tensile properties of the manufactured IN625 plate were determined in the horizontal (X) and building directions (Z). Specific locations and shapes of the specimens are shown in Figure 2. For quasi-static testing, three specimens were tested in the horizontal and building directions (top and bottom areas), respectively. The used specimen geometry is shown in Figure 2b,c. The quasi-static tensile tests were carried out in air at room temperature using a Schenck PSB100 with a 75 kN load cell and an Instron^®^ Controller 8800 (Instron^®^, Norwood, MA, USA). For strain measurement a tactile extensometer with a gauge length of 10 mm was used. The tests were carried out under strain control at a strain rate of 2.5 × 10^−4^ s^−1^.

In order to further evaluate the mechanical behavior of the investigated alloy, quasi-static compression tests at room temperature and at temperatures of 400 and 650 °C were carried out using the same servohydraulic testing system as for the tensile tests. During compression tests at different temperatures the system was supplemented with a high-temperature furnace MTS 653 (MTS Systems, Eden Prairie, MN, USA), which can reach temperatures up to 1100 °C. For specimen preparation, wire eroding was used to cut cylindrical specimens out of the as-built material along the build direction. Specimens had final dimensions of 4 mm in diameter and heights of 6 mm, according to [17]. During the experiment, the cylindrical specimens were placed on compression dies made of tungsten carbide cobalt and further positioned with a centering device for uniform force transmission. Additionally, the compression dies and specimens were lubricated with a temperature-resistant ceramic paste (LIQUI MOLY, Ulm, Germany), according to DIN 50106 [18]. The tests were carried out at a controlled speed (v_c_ = 0.0025 mm/s), and the displacements of the traverse and the forces were plotted.

## 3. Results

### 3.1. Process-Induced Microstructure and Initial Hardness

The typical microstructure of the arc-welded IN625 is presented in Figure 3; in Figure 3a, a light microscope image of the cross section perpendicular to the building direction is shown. There it can be seen that dendrites grow parallel to the building direction, and deviating dendrite growth directions cannot be found. The corresponding light microscope image along the building direction is depicted in Figure 3b, showing a microstructure consisting of columnar-oriented dendrites, mostly growing in the building direction. However, the dendrite growth direction differed significantly in local areas, mainly caused by the changing temperature gradient. Due to epitaxial growth, no single layers can be detected in the light microscope images.

Figure 4 shows a perpendicular-oriented SEM image of IN625. There, dendritic and interdendritic areas can be clearly distinguished. Furthermore, firmly embedded carbides smaller than 1 µm can be detected, causing a high grade of local deformation with additional cracks in the interdendritic area.

Due to the high density of IN625, µ-CT was not used to determine process-induced porosity. However, 2D pore analysis was carried out, and averaged results are plotted in Table 1. A total area of around 12 × 10^6^ µm^2^ was investigated. The results of the 2D analysis highlight that pores were present in IN625 and the relative density was found to be 99.7%.

Hardness profiles of the Y–Z plane from the bottom to top regions of the investigated material are presented in Figure 5. The hardness seemed to be heterogeneous, ranging from 205 to 222 HV with an average hardness value of 212 ± 8 HV. Furthermore, hardness values increased linearly with increasing build height, and the maximum hardness value was found in the near top region.

### 3.2. Mechanical Properties

For the following results, vertical specimens are declared as V_T_ (top region) and V_B_ (bottom region); for horizontal specimens, the abbreviation H is used with an index for bottom (B), middle (M), and top (T) regions (Figure 2a). The obtained averaged test results are depicted in Figure 6 and corresponding values for yield strength σ_0.2%_, ultimate tensile strength σ_UTS_, and fracture strain ε_f_ are listed in Table 2.

As can be seen, bottom specimens showed the lowest overall values for σ_0.2%_ and σ_UTS_. The maximum σ_0.2%_ was found to be 426 MPa for the horizontal specimen extracted from the middle of the component. The same was true for σ_UTS_, which was about 765 MPa. However, differences in mechanical strength between horizontally and vertically oriented specimens were only slight.

With regard to ε_f_, differences between the various specimens were more significant. At first, it can be seen that ε_f_ increased with increasing build height for both types of specimen. However, ε_f_ was generally higher for vertically oriented specimens, reaching differences of more than 10% in comparison to horizontally oriented specimens, implying that orientation to load direction influenced ε_f_ instead of σ_0.2%_ and σ_UTS_. Maximum values of ε_f_ were reached by the vertically oriented specimens, extracted from the top region, and were found to be almost 68%.

Even though tensile tests were realized, complementary compression tests were used to describe the mechanical behavior under even higher temperatures. Corresponding results of the compression tests are plotted as stress–compression curves in Figure 7.

As can be seen, the progress of the curves in the elastic and plastic regions looks nearly the same and it is evident that the mechanical strength decreased from room temperature (RT) to higher temperature levels. However, differences between 400 and 650 °C were not present. Between temperatures of 550 and 650 °C, IN625 formed a metastable phase γ″-Ni_3_(Nb,Al,Ti) leading to precipitation hardening which might explain the stable compressive strength at 650 °C [19,20]. Furthermore, no fracture was observed during compression tests up to ε_c_ = 20%. Therefore, no compressive strength could be determined. However, based on DIN 50106 [18] alternative characteristic values could be determined at different states of plastic deformation, analogous to compression strength σ_c,p,0.2_.

For the following, compression strength σ_c,p,2_ and σ_c,p,10_, corresponding to 2% and 10% plastic deformation were introduced and results are plotted in Figure 8. Furthermore, absolute values are listed in Table 3. As mentioned above, compressive strength decreased from RT to 400 °C, however, at higher temperatures a stable level of compressive strength, independent of the degree of plastic deformation, could be determined. In particular, σ_c,p,0.2_ at RT was found to be 427 MPa which subsequently decreased to around 318 MPa at higher temperature. The same tendency could be detected for σ_c,p,2_ and σ_c,p,10_.

### 3.3. Fractographic Analysis

A representative fracture surface after the tensile test for IN625 is shown in Figure 9. In general, a high degree of deformation in the form of high elongation at fracture were detected and failure occurred under an angle of 45°. Additionally, shear zones developed at the border areas, indicating a ductile fracture. In the corresponding magnification image, the fracture surface shows a honeycomb structure with a high number of deformed dimples, corresponding to a transgranular ductile fracture. Inside, a certain number of dimples small pores were identified, which locally supported necking during the tensile test. Differences in the fractographic analysis between vertically and horizontally oriented tensile specimens were not visible.

## 4. Discussion

The microstructure of nickel-based superalloys was reported to be sensitive to the applied process, which translates to a dependency on the solidification conditions and the thermodynamic equilibrium. In the shape metal deposition (SMD) of IN718 [8] the microstructure was hierarchical, which consisted of a matrix and precipitates and then substructures rich in Nb. The selective-laser-melted (SLM) IN625 [5] showed clear fusion lines at the heat-affected zones under the SEM. Such a morphology was not present in the current study since the structure was arc deposited with a much larger heat focus zone, leading to a more homogenous microstructure with less observable discontinuities. The cellular structure in the case of selective laser melting was reported to mainly consist of Nb, which will be worth investigating in a further study. How the Nb will be composed in the case of arc deposition will be investigated in the next study of this project, where variation of parameters is planned. Studies in SLM have highlighted that the nature of this composition and its scale of distribution are sensitive to the applied laser power.

Gonzalez et al. [21] explored the evolution of columnar grains in the build direction following the cooling gradient in electron beam melting. Dimples on the tensile-tested specimens indicated a ductile fracture similar to what is presented in Figure 9. That remains different from binder jetting, which showed a location of unsintered powder on the fracture surface that indicated a tendency towards brittle fracture. Heterogeneities were reported to be responsible for fatigue damage and failure [22]. Similar micro-failures are identified in Figure 4. Although very high localized thermal gradients dominated solidification in additively manufactured IN625, the long-crack propagation rate was found comparable with wrought IN625 [23].

Zhang et al. [24] attempted to improve the strength of laser-melted IN625 by carbon, which resulted in 20% improvement due to grain refinement and pinning grain boundary. On the other hand, ductility decreased. Further extension of the current work designates corrosion resistance for industrial application, which Cabrini et al. [25] attempted to enhance by heat treatment at 980 °C for 32 min. Hot isostatic pressing increased the fatigue limit at 1E8 cycles by 87% to 620 MPa, which decreased to 540 MPa during testing in saltwater [26]. Current research suggests that authors compare these results to the wire-arc-deposited IN625 in this study.

The profile of the presented X-ray porosity analysis here was dominated by the challenges in penetrating the high-density IN625. The authors tried to ensure the coherency and the integrity of the study by analyzing the porosity of IN625 based on optical methods while isolating the porosity results, class-wise. The authors concluded that future studies with in-depth analyses of porosity using various methods such as building virtual volumes of porosity in IN625 by ultrasonic waves would be necessary if a solid comparison is to be made.

The hierarchy of the microstructure of the alloying systems significantly influenced the macroscale deformation in the quasi-static tests. The matrix-precipitate composition allowed significant plastic deformation and necking due to enhanced slipping in the FCC (face-centered cubic) lattices of the Ni matrix. If the flow strengths of IN625 at room and elevated temperatures are compared, it would be realized that around 30% of the flow strength is diminished in the case of elevated temperatures.

## 5. Conclusions

In the current study the authors highlighted the potential of fusion-based additive manufacturing processes in manufacturing high-temperature aerospace materials. The porosity prone structures were investigated with light microscopy for IN625 as they proved too challenging to be penetrated by X-rays. The average hardness of 212 HV was determined, however, and heterogeneous hardness distribution with maximum hardness in the top region was detected. This tendency was further confirmed based on quasi-static tensile tests, since specimens from the top region showed higher yield and tensile strength when compared to the bottom region. Furthermore, differences became more significant when comparing fracture strain, which was found to be almost 68% for specimens in the top region. The ductile fracture was proved by the observation of dimples on fracture surfaces. The heterogeneities of the microstructural precipitates are believed to cause local failures that are expected to be activated during fatigue testing and cause multiple crack initiations. 

Compression tests at elevated temperatures proved the good high-temperature properties of arc-welded IN625 because of thermal stability.

The results highlight potentially interesting questions about the role of this microstructure in a cyclic creep study. Pre- and post-EBSD analysis of the planned cyclic creep tests will reveal the role of texture induced by arc metal deposition and electron beam melting and if recrystallization plays a role in fatigue strength at elevated temperatures.

## Figures and Tables

**Figure 1 materials-12-03690-f001:**
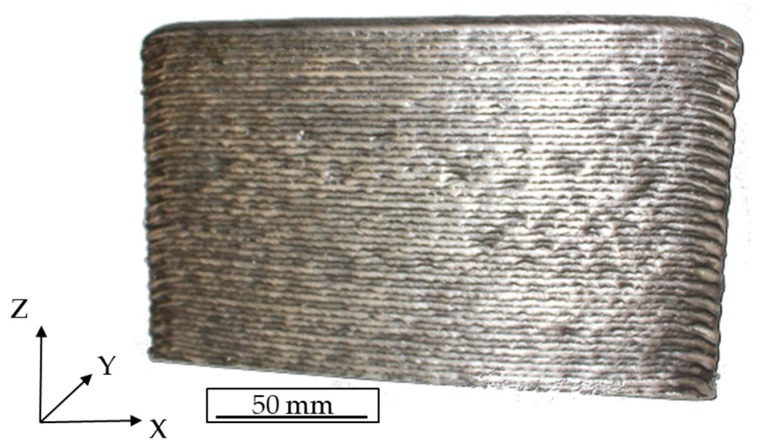
As-deposited arc-welded nickel-based superalloy Inconel 625 (IN625).

**Figure 2 materials-12-03690-f002:**
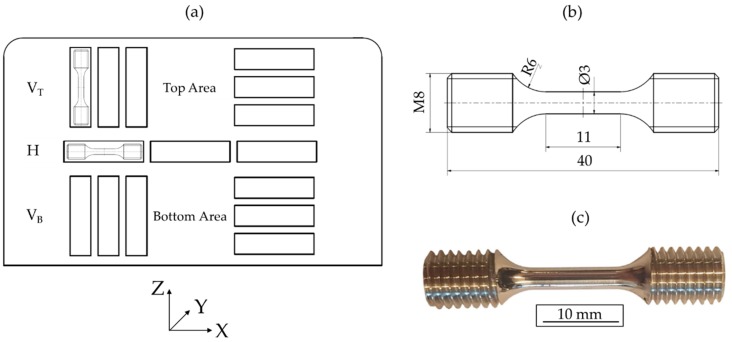
Schematic configuration of tensile specimens: (**a**) positions, (**b**) dimensions (in mm), and (**c**) final geometry.

**Figure 3 materials-12-03690-f003:**
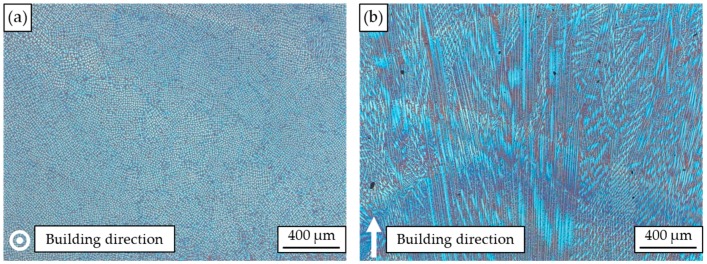
Optical microscope images of arc-welded IN625: cross section (**a**) perpendicular and (**b**) parallel to the building direction.

**Figure 4 materials-12-03690-f004:**
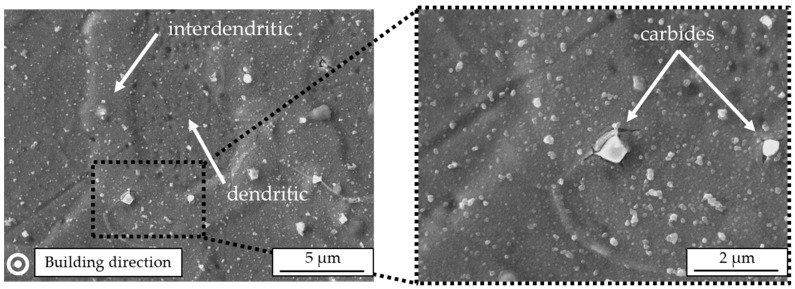
Scanning electron microscope images of arc-welded IN625 perpendicular to the building direction.

**Figure 5 materials-12-03690-f005:**
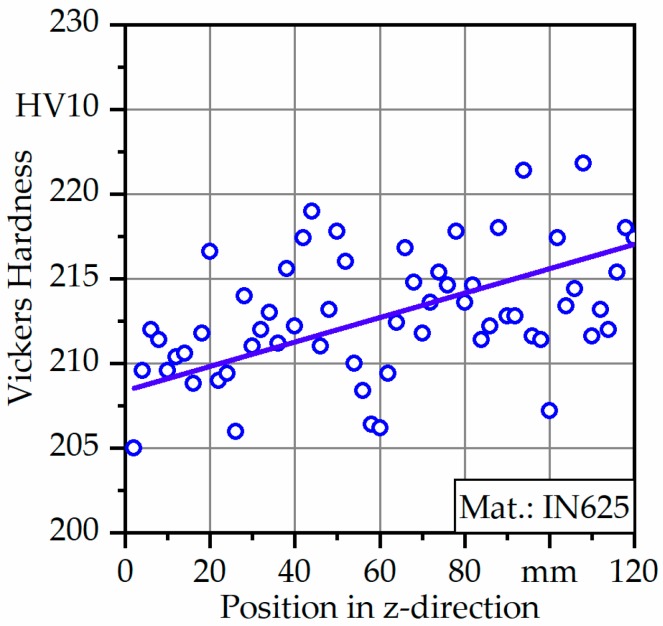
Hardness profile of IN625 along the building direction in a cross section (Y–Z plane).

**Figure 6 materials-12-03690-f006:**
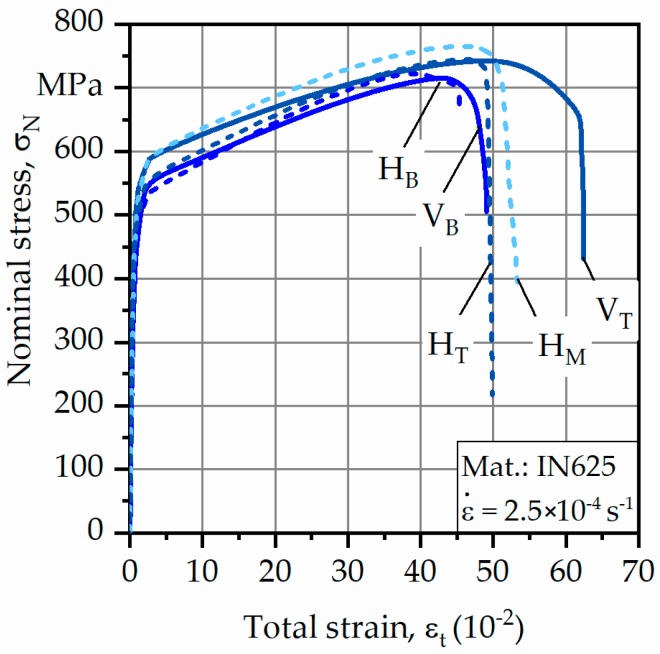
Averaged results of tensile tests.

**Figure 7 materials-12-03690-f007:**
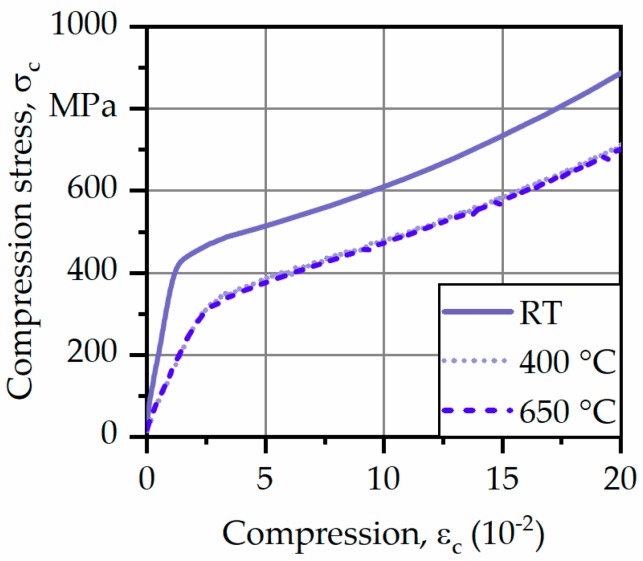
Averaged results of compression tests. RT: room temperature.

**Figure 8 materials-12-03690-f008:**
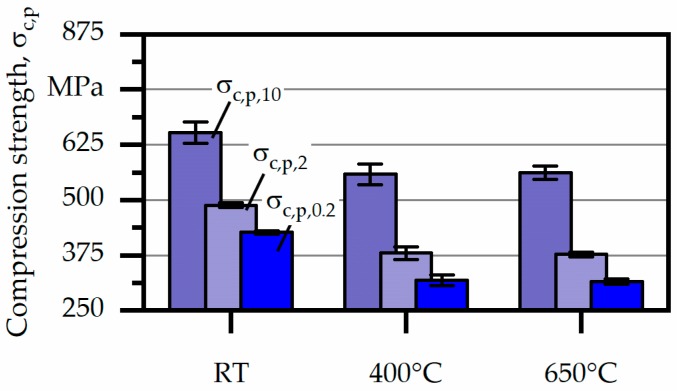
Compression strength σ_c,p_ for the investigated materials at different temperatures.

**Figure 9 materials-12-03690-f009:**
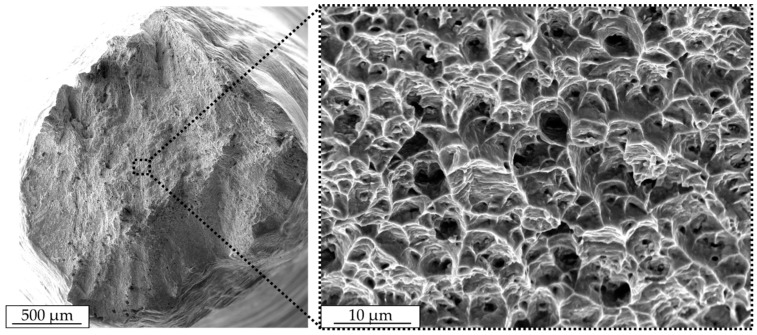
Fracture surface with the corresponding magnification of IN625 after a tensile test in the scanning electron microscope.

**Table 1 materials-12-03690-t001:** 2D pore analysis of the investigated material.

Material	Area of Pores	Relative Density
IN625	31,682 µm^2^	99.74% ± 0.02%

**Table 2 materials-12-03690-t002:** Averaged results of quasi-static tensile tests.

Material	IN625 H_B_	IN625 H_M_	IN625 H_T_	IN625 V_B_	IN625 V_T_
σ_0.2%_ (MPa)	343 ± 22	426 ± 1	417 ± 7	387 ± 4	421 ± 13
σ_UTS_ (MPa)	722 ± 37	765 ± 9	744 ± 29	715 ± 28	742 ± 1
ε_f_ (10^−2^)	52.3 ± 6.8	55.6 ± 2.9	56.7 ± 5.4	60.8 ± 8.9	67.9 ± 5.8

**Table 3 materials-12-03690-t003:** Averaged results of quasi-static compression tests.

Material	RT	400 °C	650 °C
σ_c,p,0.2_ (MPa)	427 ± 4	318 ± 12	316 ± 6
σ_c,p,2_ (MPa)	488 ± 5	380 ± 14	377 ± 5
σ_c,p,10_ (MPa)	652 ± 24	558 ± 24	562 ± 15

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
