# Peer review of "Mechanical and Microstructural Characterization of Arc-Welded Inconel 625 Alloy"

_materials, 2019, doi:10.3390/ma12223690_

Round 1

Reviewer 1 Report

The authors report an interesting research work on the realization of IN625-based samples by using the wire arc additive manufacturing technique. The overall quality of the manuscript is of relevance, highlighting an appropriate design and description of both methods and results. However, the reviewer suggests to reconsider the article after major revision for the following reason:

Line 65, the authors state that no further details are available. The reviewer suggests to provide more information about the process parameters in order to make the experiments replicable, which is of crucial importance for both academic and indsustrial research and development. Moreover, the authors should also provide the reason why they chose such values. A further investigation could also concern the influcence of varying such parameters on the mechanical properties of the samples.

Author Response

Author's reply is attached.

Reviewer 2 Report

In this work, through performing material performance tests under nickel-based superalloy Inconel 625 (IN625) manufactured by arc-based AM technique, the authors highlight the potential of fusion-based additive manufacturing processes in manufacturing high-temperature aerospace materials, and utilized the experimental data and scanning electron microscope images of arc-welded IN625 for model validation and comparison. Results of the current work is of interest for the field, however, the following comments should be addressed for further consideration.

In Fig. 7, the phenomenon “differences between 400 and 650°C are not present” should be discussed more; In Section 4, the statement "How the Nb will be composed in case of arc deposition since studies in SLM highlighted that the nature of this composition and its scale of distribution is sensitive to the applied laser power" needs to be clarified. In Section 3.5, any differences on the fractographic analysis of vertical/horizontal specimens? The conclusion sections should be refined and shortened and focused on the new points of current work.

Author Response

Author's reply is attached.

Round 2

Reviewer 1 Report

The authors improved the overall quality of the manuscript. However, since it is not possible to provide any further information about the process parameters, the reviewer suggests to indicate the producer of the samples at the beginning of Section 2 (Line 60 of the revised manuscript).

Author Response

We would like to thank the reviewer for the valuable comments. The manuscript has been modified in order to address this point.